# Minimax-optimal Inference from Partial Rankings

**Bruce Hajek**
UIUC
b-hajek@illinois.edu

**Sewoong Oh**
UIUC
swoh@illinois.edu

**Jiaming Xu**
UIUC
jxu18@illinois.edu

## Abstract

This paper studies the problem of rank aggregation under the Plackett-Luce model. The goal is to infer a global ranking and related scores of the items, based on partial rankings provided by multiple users over multiple subsets of items. A question of particular interest is how to optimally assign items to users for ranking and how many item assignments are needed to achieve a target estimation error. Without any assumptions on how the items are assigned to users, we derive an oracle lower bound and the Cramér-Rao lower bound of the estimation error. We prove an upper bound on the estimation error achieved by the maximum likelihood estimator, and show that both the upper bound and the Cramér-Rao lower bound inversely depend on the spectral gap of the Laplacian of an appropriately defined comparison graph. Since random comparison graphs are known to have large spectral gaps, this suggests the use of random assignments when we have the control. Precisely, the matching oracle lower bound and the upper bound on the estimation error imply that the maximum likelihood estimator together with a random assignment is minimax-optimal up to a logarithmic factor. We further analyze a popular rank-breaking scheme that decompose partial rankings into pairwise comparisons. We show that even if one applies the mismatched maximum likelihood estimator that assumes independence (on pairwise comparisons that are now dependent due to rank-breaking), minimax optimal performance is still achieved up to a logarithmic factor.

## 1 Introduction

Given a set of individual preferences from multiple decision makers or judges, we address the problem of computing a consensus ranking that best represents the preference of the population collectively. This problem, known as rank aggregation, has received much attention across various disciplines including statistics, psychology, sociology, and computer science, and has found numerous applications including elections, sports, information retrieval, transportation, and marketing [1, 2, 3, 4]. While consistency of various rank aggregation algorithms has been studied when a growing number of sampled partial preferences is observed over a fixed number of items [5, 6], little is known in the high-dimensional setting where the number of items and number of observed partial rankings scale simultaneously, which arises in many modern datasets. Inference becomes even more challenging when each individual provides limited information. For example, in the well known Netflix challenge dataset, 480,189 users submitted ratings on 17,770 movies, but on average a user rated only 209 movies. To pursue a rigorous study in the high-dimensional setting, we assume that users provide partial rankings over subsets of items generated according to the popular Plackett-Luce (PL) model [7] from some hidden preference vector over all the items and are interested in estimating the preference vector (see Definition 1).

Intuitively, inference becomes harder when few users are available, or each user is assigned few items to rank, meaning fewer observations. The first goal of this paper is to quantify the number of item assignments needed to achieve a target estimation error. Secondly, in many practical scenarios such as crowdsourcing, the systems have the control over the item assignment. For such systems, a

natural question of interest is how to optimally assign the items for a given budget on the total number of item assignments. Thirdly, a common approach in practice to deal with partial rankings is to break them into pairwise comparisons and apply the state-of-the-art rank aggregation methods specialized for pairwise comparisons [8, 9]. It is of both theoretical and practical interest to understand how much the performance degrades when rank breaking schemes are used.

**Notation.** For any set $S$, let $|S|$ denote its cardinality. Let $s_1^n = \{s_1, \ldots, s_n\}$ denote a set with $n$ elements. For any positive integer $N$, let $[N] = \{1, \ldots, N\}$. We use standard big $O$ notations, e.g., for any sequences $\{a_n\}$ and $\{b_n\}$, $a_n = \Theta(b_n)$ if there is an absolute constant $C > 0$ such that $1/C \leq a_n/b_n \leq C$. For a partial ranking $\sigma$ over $S$, i.e., $\sigma$ is a mapping from $[|S|]$ to $S$, let $\sigma^{-1}$ denote the inverse mapping. All logarithms are natural unless the base is explicitly specified. We say a sequence of events $\{A_n\}$ holds with high probability if $\mathbb{P}[A_n] \geq 1 - c_1 n^{-c_2}$ for two positive constants $c_1, c_2$.

## 1.1 Problem setup

We describe our model in the context of recommender systems, but it is applicable to other systems with partial rankings. Consider a recommender system with $m$ users indexed by $[m]$ and $n$ items indexed by $[n]$. For each item $i \in [n]$, there is a hidden parameter $\theta_i^*$ measuring the underlying preference. Each user $j$, independent of everyone else, randomly generates a partial ranking $\sigma_j$ over a subset of items $S_j \subseteq [n]$ according to the PL model with the underlying preference vector $\theta^* = (\theta_1^*, \ldots, \theta_n^*)$.

**Definition 1** (PL model). A partial ranking $\sigma : [|S|] \to S$ is generated from $\{\theta_i^*, i \in S\}$ under the PL model in two steps: (1) independently assign each item $i \in S$ an unobserved value $X_i$, exponentially distributed with mean $e^{-\theta_i^*}$; (2) select $\sigma$ so that $X_{\sigma(1)} \leq X_{\sigma(2)} \leq \cdots \leq X_{\sigma(|S|)}$.

The PL model can be equivalently described in the following sequential manner. To generate a partial ranking $\sigma$, first select $\sigma(1)$ in $S$ randomly from the distribution $e^{\theta_i^*}/\left(\sum_{i' \in S} e^{\theta_{i'}^*}\right)$; secondly, select $\sigma(2)$ in $S \setminus \{\sigma(1)\}$ with the probability distribution $e^{\theta_i^*}/\left(\sum_{i' \in S \setminus \{\sigma(1)\}} e^{\theta_{i'}^*}\right)$; continue the process in the same fashion until all the items in $S$ are assigned. The PL model is a special case of the following class of models.

**Definition 2** (Thurstone model, or random utility model (RUM) ). A partial ranking $\sigma : [|S|] \to S$ is generated from $\{\theta_i^*, i \in S\}$ under the Thurstone model for a given CDF $F$ in two steps: (1) independently assign each item $i \in S$ an unobserved utility $U_i$, with CDF $F(c - \theta_i^*)$; (2) select $\sigma$ so that $U_{\sigma(1)} \geq U_{\sigma(2)} \geq \cdots \geq U_{\sigma(|S|)}$.

To recover the PL model from the Thurstone model, take $F$ to be the CDF for the standard Gumbel distribution: $F(c) = e^{-(e^{-c})}$. Equivalently, take $F$ to be the CDF of $-\log(X)$ such that $X$ has the exponential distribution with mean one. For this choice of $F$, the utility $U_i$ having CDF $F(c - \theta_i^*)$, is equivalent to $U_i = -\log(X_i)$ such that $X_i$ is exponentially distributed with mean $e^{-\theta_i^*}$. The corresponding partial permutation $\sigma$ is such that $X_{\sigma(1)} \leq X_{\sigma(2)} \leq \cdots \leq X_{\sigma(|S|)}$, or equivalently, $U_{\sigma(1)} \geq U_{\sigma(2)} \geq \cdots \geq U_{\sigma(|S|)}$. (Note the opposite ordering of $X$'s and $U$'s.)

Given the observation of all partial rankings $\{\sigma_j\}_{j \in [m]}$ over the subsets $\{S_j\}_{j \in [m]}$ of items, the task is to infer the underlying preference vector $\theta^*$. For the PL model, and more generally for the Thurstone model, we see that $\theta^*$ and $\theta^* + a\mathbf{1}$ for any $a \in \mathbb{R}$ are statistically indistinguishable, where $\mathbf{1}$ is an all-ones vector. Indeed, under our model, the preference vector $\theta^*$ is the equivalence class $[\theta^*] = \{\theta : \exists a \in \mathbb{R}, \theta = \theta^* + a\mathbf{1}\}$. To get a unique representation of the equivalence class, we assume $\sum_{i=1}^n \theta_i^* = 0$. Then the space of all possible preference vectors is given by $\Theta = \{\theta \in \mathbb{R}^n : \sum_{i=1}^n \theta_i = 0\}$. Moreover, if $\theta_i^* - \theta_{i'}^*$ becomes arbitrarily large for all $i' \neq i$, then with high probability item $i$ is ranked higher than any other item $i'$ and there is no way to estimate $\theta_i$ to any accuracy. Therefore, we further put the constraint that $\theta^* \in [-b, b]^n$ for some $b \in \mathbb{R}$ and define $\Theta_b = \Theta \cap [-b, b]^n$. The parameter $b$ characterizes the dynamic range of the underlying preference. In this paper, we assume $b$ is a fixed constant. As observed in [10], if $b$ were scaled with $n$, then it would be easy to rank items with high preference versus items with low preference and one can focus on ranking items with close preference.

We denote the number of items assigned to user $j$ by $k_j := |S_j|$ and the average number of assigned items per use by $k = \frac{1}{m}\sum_{j=1}^{m} k_j$; parameter $k$ may scale with $n$ in this paper. We consider two scenarios for generating the subsets $\{S_j\}_{j=1}^{m}$: the random item assignment case where the $S_j$'s are chosen independently and uniformly at random from all possible subsets of $[n]$ with sizes given by the $k_j$'s, and the deterministic item assignment case where the $S_j$'s are chosen deterministically.

Our main results depend on the structure of a weighted undirected graph $G$ defined as follows.

**Definition 3** (Comparison graph $G$). Each item $i \in [n]$ corresponds to a vertex $i \in [n]$. For any pair of vertices $i, i'$, there is a weighted edge between them if there exists a user who ranks both items $i$ and $i'$; the weight equals $\sum_{j:i,i' \in S_j} \frac{1}{k_j-1}$.

Let $A$ denote the weighted adjacency matrix of $G$. Let $d_i = \sum_j A_{ij}$, so $d_i$ is the number of users who rank item $i$, and without loss of generality assume $d_1 \le d_2 \le \cdots \le d_n$. Let $D$ denote the $n \times n$ diagonal matrix formed by $\{d_i, i \in [n]\}$ and define the graph Laplacian $L$ as $L = D - A$. Observe that $L$ is positive semi-definite and the smallest eigenvalue of $L$ is zero with the corresponding eigenvector given by the normalized all-one vector. Let $0 = \lambda_1 \le \lambda_2 \le \cdots \le \lambda_n$ denote the eigenvalues of $L$ in ascending order.

**Summary of main results.** Theorem 1 gives a lower bound for the estimation error that scales as $\sum_{i=2}^{n} \frac{1}{d_i}$. The lower bound is derived based on a genie-argument and holds for both the PL model and the more general Thurstone model. Theorem 2 shows that the Cramér-Rao lower bound scales as $\sum_{i=2}^{n} \frac{1}{\lambda_i}$. Theorem 3 gives an upper bound for the squared error of the maximum likelihood (ML) estimator that scales as $\frac{mk \log n}{(\lambda_2 - \sqrt{\lambda_n})^2}$. Under the full rank breaking scheme that decomposes a $k$-way comparison into $\binom{k}{2}$ pairwise comparisons, Theorem 4 gives an upper bound that scales as $\frac{mk \log n}{\lambda_2^2}$. If the comparison graph is an expander graph, i.e., $\lambda_2 \sim \lambda_n$ and $mk = \Omega(n \log n)$, our lower and upper bounds match up to a $\log n$ factor. This follows from the fact that $\sum_i \lambda_i = \sum_i d_i = mk$, and for expanders $mk = \Theta(n\lambda_2)$. Since the Erdős-Rényi random graph is an expander graph with high probability for average degree larger than $\log n$, when the system is allowed to choose the item assignment, we propose a random assignment scheme under which the items for each user are chosen *independently and uniformly at random*. It follows from Theorem 1 that $mk = \Omega(n)$ is *necessary* for any item assignment scheme to reliably infer the underlying preference vector, while our upper bounds imply that $mk = \Omega(n \log n)$ is *sufficient* with the random assignment scheme and can be achieved by either the ML estimator or the full rank breaking or the independence-preserving breaking that decompose a $k$-way comparison into $\lfloor k/2 \rfloor$ non-intersecting pairwise comparisons, proving that rank breaking schemes are also nearly optimal.

## 1.2 Related Work

There is a vast literature on rank aggregation, and here we can only hope to cover a fraction of them we see most relevant. In this paper, we study a statistical learning approach, assuming the observed ranking data is generated from a probabilistic model. Various probabilistic models on permutations have been studied in the ranking literature (see, e.g., [11, 12]). A nonparametric approach to modeling distributions over rankings using sparse representations has been studied in [13]. Most of the parametric models fall into one of the following three categories: noisy comparison model, distance based model, and random utility model. The noisy comparison model assumes that there is an underlying true ranking over $n$ items, and each user independently gives a pairwise comparison which agrees with the true ranking with probability $p > 1/2$. It is shown in [14] that $O(n \log n)$ pairwise comparisons, when chosen adaptively, are sufficient for accurately estimating the true ranking.

The Mallows model is a distance-based model, which randomly generates a full ranking $\sigma$ over $n$ items from some underlying true ranking $\sigma^*$ with probability proportional to $e^{-\beta d(\sigma,\sigma^*)}$, where $\beta$ is a fixed spread parameter and $d(\cdot, \cdot)$ can be any permutation distance such as the Kemeny distance. It is shown in [14] that the true ranking $\sigma^*$ can be estimated accurately given $O(\log n)$ independent full rankings generated under the Mallows model with the Kemeny distance.

In this paper, we study a special case of random utility models (RUMs) known as the Plackett-Luce (PL) model. It is shown in [7] that the likelihood function under the PL model is concave and the ML estimator can be efficiently found using a minorization-maximization (MM) algorithm which is

a variation of the general EM algorithm. We give an upper bound on the error achieved by such an ML estimator, and prove that this is matched by a lower bound. The lower bound is derived by comparing to an oracle estimator which observes the random utilities of RUM directly. The Bradley-Terry (BT) model is the special case of the PL model where we only observe pairwise comparisons. For the BT model, [10] proposes RankCentrality algorithm based on the stationary distribution of a random walk over a suitably defined comparison graph and shows $\Omega(n\mathsf{poly}(\log n))$ randomly chosen pairwise comparisons are sufficient to accurately estimate the underlying parameters; one corollary of our result is a matching performance guarantee for the ML estimator under the BT model. More recently, [15] analyzed various algorithms including RankCentrality and the ML estimator under a general, not necessarily uniform, sampling scheme.

In a PL model with priors, MAP inference becomes computationally challenging. Instead, an efficient message-passing algorithm is proposed in [16] to approximate the MAP estimate. For a more general family of random utility models, Soufiani et al. in [17, 18] give a sufficient condition under which the likelihood function is concave, and propose a Monte-Carlo EM algorithm to compute the ML estimator for general RUMs. More recently in [8, 9], the generalized method of moments together with the rank-breaking is applied to estimate the parameters of the PL model and the random utility model when the data consists of full rankings.

## 2 Main results

In this section, we present our theoretical findings and numerical experiments.

### 2.1 Oracle lower bound

In this section, we derive an oracle lower bound for any estimator of $\theta^*$. The lower bound is constructed by considering an oracle who reveals all the hidden scores in the PL model as side information and holds for the general Thurstone models.

**Theorem 1.** *Suppose $\sigma_1^m$ are generated from the Thurstone model for some CDF F. For any estimator $\widehat{\theta}$,*

$$\inf_{\widehat{\theta}} \sup_{\theta^* \in \Theta_b} E[||\widehat{\theta} - \theta^*||_2^2] \geq \frac{1}{2I(\mu) + \frac{2\pi^2}{b^2(d_1 + d_2)}} \sum_{i=2}^{n} \frac{1}{d_i} \geq \frac{1}{2I(\mu) + \frac{2\pi^2}{b^2(d_1 + d_2)}} \frac{(n-1)^2}{mk},$$

*where $\mu$ is the probability density function of F, i.e., $\mu = F'$ and $I(\mu) = \int \frac{\left(\mu'(x)\right)^2}{\mu(x)} dx$; the second inequality follows from the Jensen's inequality. For the PL model, which is a special case of the Thurstone models with F being the standard Gumbel distribution, $I(\mu) = 1$.*

Theorem 1 shows that the oracle lower bound scales as $\sum_{i=2}^{n} \frac{1}{d_i}$. We remark that the summation begins with $1/d_2$. This makes some sense, in view of the fact that the parameters $\theta_i^*$ need to sum to zero. For example, if $d_1$ is a moderate value and all the other $d_i$'s are very large, then with the hidden scores as side information, we may be able to accurately estimate $\theta_i^*$ for $i \neq 1$ and therefore accurately estimate $\theta_1^*$. The oracle lower bound also depends on the dynamic range $b$ and is tight for $b = 0$, because a trivial estimator that always outputs the all-zero vector achieves the lower bound.

**Comparison to previous work** Theorem 1 implies that $mk = \Omega(n)$ is necessary for any item assignment scheme to reliably infer $\theta^*$, i.e., ensuring $E[||\widehat{\theta} - \theta^*||_2^2] = o(n)$. It provides the first converse result on inferring the parameter vector under the general Thurstone models to our knowledge. For the Bradley-Terry model, which is a special case of the PL model where all the partial rankings reduce to the pairwise comparisons, i.e., $k = 2$, it is shown in [10] that $m = \Omega(n)$ is necessary for the random item assignment scheme to achieve the reliable inference based on the information-theoretic argument. In contrast, our converse result is derived based on the Bayesian Cramé-Rao lower bound [19], applies to the general models with any item assignment, and is considerably tighter if $d_i$'s are of different orders.

### 2.2 Cramér-Rao lower bound

In this section, we derive the Cramér-Rao lower bound for any unbiased estimator of $\theta^*$.

**Theorem 2.** *Let $k_{\max} = \max_{j \in [m]} k_j$ and $\mathcal{U}$ denote the set of all unbiased estimators of $\theta^*$, i.e., $\widehat{\theta} \in \mathcal{U}$ if and only if $\mathbb{E}[\widehat{\theta}|\theta^* = \theta] = \theta, \forall \theta \in \Theta_b$. If $b > 0$, then*

$$\inf_{\widehat{\theta} \in \mathcal{U}} \sup_{\theta^* \in \Theta_b} \mathbb{E}[\|\widehat{\theta} - \theta^*\|_2^2] \geq \left(1 - \frac{1}{k_{\max}} \sum_{\ell=1}^{k_{\max}} \frac{1}{\ell}\right)^{-1} \sum_{i=2}^{n} \frac{1}{\lambda_i} \geq \left(1 - \frac{1}{k_{\max}} \sum_{\ell=1}^{k_{\max}} \frac{1}{\ell}\right)^{-1} \frac{(n-1)^2}{mk},$$

*where the second inequality follows from the Jensen's inequality.*

The Cramér-Rao lower bound scales as $\sum_{i=2}^{n} \frac{1}{\lambda_i}$. When $G$ is disconnected, i.e., all the items can be partitioned into two groups such that no user ever compares an item in one group with an item in the other group, $\lambda_2 = 0$ and the Cramér-Rao lower bound is infinity, which is valid (and of course tight) because there is no basis for gauging any item in one connected component with respect to any item in the other connected component and the accurate inference is impossible for any estimator. Although the Cramér-Rao lower bound only holds for any unbiased estimator, we suspect that a lower bound with the same scaling holds for any estimator, but we do not have a proof.

### 2.3 ML upper bound

In this section, we study the ML estimator based on the partial rankings. The ML estimator of $\theta^*$ is defined as $\widehat{\theta}_{\mathsf{ML}} \in \arg\max_{\theta \in \Theta_b} \mathcal{L}(\theta)$, where $\mathcal{L}(\theta)$ is the log likelihood function given by

$$\mathcal{L}(\theta) = \log \mathbb{P}_\theta[\sigma_1^m] = \sum_{j=1}^{m} \sum_{\ell=1}^{k_j-1} \left[\theta_{\sigma_j(\ell)} - \log\left(\exp(\theta_{\sigma_j(\ell)}) + \cdots + \exp(\theta_{\sigma_j(k_j)})\right)\right]. \quad (1)$$

As observed in [7], $\mathcal{L}(\theta)$ is concave in $\theta$ and thus the ML estimator can be efficiently computed either via the gradient descent method or the EM type algorithms.

The following theorem gives an upper bound on the error rates inversely dependent on $\lambda_2$. Intuitively, by the well-known Cheeger's inequality, if the spectral gap $\lambda_2$ becomes larger, then there are more edges across any bi-partition of $G$, meaning more pairwise comparisons are available between any bi-partition of movies, and therefore $\theta^*$ can be estimated more accurately.

**Theorem 3.** *Assume $\lambda_n \geq C \log n$ for a sufficiently large constant $C$ in the case with $k > 2$. Then with high probability,*

$$\|\widehat{\theta}_{\mathsf{ML}} - \theta^*\|_2 \leq \begin{cases} 4(1 + e^{2b})^2 \lambda_2^{-1} \sqrt{m \log n} & \text{If } k = 2, \\ \frac{8 e^{4b} \sqrt{2mk \log n}}{\lambda_2 - 16 e^{2b} \sqrt{\lambda_n \log n}} & \text{If } k > 2. \end{cases}$$

We compare the above upper bound with the Cramér-Rao lower bound given by Theorem 2. Notice that $\sum_{i=1}^{n} \lambda_i = mk$ and $\lambda_1 = 0$. Therefore, $\frac{mk}{\lambda_2^2} \geq \sum_{i=2}^{n} \frac{1}{\lambda_i}$ and the upper bound is always larger than the Cramér-Rao lower bound. When the comparison graph $G$ is an expander and $mk = \Omega(n \log n)$, by the well-known Cheeger's inequality, $\lambda_2 \sim \lambda_n = \Omega(\log n)$, the upper bound is only larger than the Cramér-Rao lower bound by a logarithmic factor. In particular, with the random item assignment scheme, we show that $\lambda_2, \lambda_n \sim \frac{mk}{n}$ if $mk \geq C \log n$ and as a corollary of Theorem 3, $mk = \Omega(n \log n)$ is sufficient to ensure $\|\widehat{\theta}_{\mathsf{ML}} - \theta^*\|_2 = o(\sqrt{n})$, proving the random item assignment scheme with the ML estimation is minimax-optimal up to a $\log n$ factor.

**Corollary 1.** *Suppose $S_1^m$ are chosen independently and uniformly at random among all possible subsets of $[n]$. Then there exists a positive constant $C > 0$ such that if $m \geq Cn \log n$ when $k = 2$ and $mk \geq C e^{2b} \log n$ when $k > 2$, then with high probability*

$$\|\widehat{\theta}_{\mathsf{ML}} - \theta^*\|_2 \leq \begin{cases} 4(1 + e^{2b})^2 \sqrt{\frac{n^2 \log n}{m}}, & \text{if } k = 2, \\ 32 e^{4b} \sqrt{\frac{2n^2 \log n}{mk}}, & \text{if } k > 2. \end{cases}$$

**Comparison to previous work**   Theorem 3 provides the first finite-sample error rates for inferring the parameter vector under the PL model to our knowledge. For the Bradley-Terry model, which is a special case of the PL model with $k = 2$, [10] derived the similar performance guarantee by analyzing the rank centrality algorithm and the ML estimator. More recently, [15] extended the results to the non-uniform sampling scheme of item pairs, but the performance guarantees obtained when specialized to the uniform sampling scheme require at least $m = \Omega(n^4 \log n)$ to ensure $\|\widehat{\theta} - \theta^*\|_2 = o(\sqrt{n})$, while our results only require $m = \Omega(n \log n)$.

## 2.4 Rank breaking upper bound

In this section, we study two rank-breaking schemes which decompose partial rankings into pairwise comparisons.

**Definition 4.** Given a partial ranking $\sigma$ over the subset $S \subset [n]$ of size $k$, the independence-preserving breaking scheme (IB) breaks $\sigma$ into $\lfloor k/2 \rfloor$ non-intersecting pairwise comparisons of form $\{i_t, i'_t, y_t\}_{t=1}^{\lfloor k/2 \rfloor}$ such that $\{i_s, i'_s\} \cap \{i_t, i'_t\} = \emptyset$ for any $s \neq t$ and $y_t = 1$ if $\sigma^{-1}(i_t) < \sigma^{-1}(i'_t)$ and 0 otherwise. The random IB chooses $\{i_t, i'_t\}_{t=1}^{\lfloor k/2 \rfloor}$ uniformly at random among all possibilities.

If $\sigma$ is generated under the PL model, then the IB breaks $\sigma$ into independent pairwise comparisons generated under the PL model. Hence, we can first break partial rankings $\sigma_1^m$ into independent pairwise comparisons using the random IB and then apply the ML estimator on the generated pairwise comparisons with the constraint that $\theta \in \Theta_b$, denoted by $\widehat{\theta}_{\mathsf{IB}}$. Under the random assignment scheme, as a corollary of Theorem 3, $mk = \Omega(n \log n)$ is sufficient to ensure $\|\widehat{\theta}_{\mathsf{IB}} - \theta^*\|_2 = o(\sqrt{n})$, proving the random item assignment scheme with the random IB is minimax-optimal up to a $\log n$ factor in view of the oracle lower bound in Theorem 1.

**Corollary 2.** *Suppose $S_1^m$ are chosen independently and uniformly at random among all possible subsets of $[n]$ with size $k$. There exists a positive constant $C > 0$ such that if $mk \geq Cn \log n$, then with high probability,*

$$\|\widehat{\theta}_{\mathsf{IB}} - \theta^*\|_2 \leq 4(1 + e^{2b})^2 \sqrt{\frac{2n^2 \log n}{mk}}.$$

**Definition 5.** Given a partial ranking $\sigma$ over the subset $S \subset [n]$ of size $k$, the full breaking scheme (FB) breaks $\sigma$ into all $\binom{k}{2}$ possible pairwise comparisons of form $\{i_t, i'_t, y_t\}_{t=1}^{\binom{k}{2}}$ such that $y_t = 1$ if $\sigma^{-1}(i_t) < \sigma^{-1}(i'_t)$ and 0 otherwise.

If $\sigma$ is generated under the PL model, then the FB breaks $\sigma$ into pairwise comparisons which are not independently generated under the PL model. We pretend the pairwise comparisons induced from the full breaking are all independent and maximize the weighted log likelihood function given by

$$\mathcal{L}(\theta) = \sum_{j=1}^m \frac{1}{2(k_j - 1)} \sum_{i, i' \in S_j} \left( \theta_i \mathbb{I}_{\{\sigma_j^{-1}(i) < \sigma_j^{-1}(i')\}} + \theta_{i'} \mathbb{I}_{\{\sigma_j^{-1}(i) > \sigma_j^{-1}(i')\}} - \log\left(e^{\theta_i} + e^{\theta_{i'}}\right) \right) \tag{2}$$

with the constraint that $\theta \in \Theta_b$. Let $\widehat{\theta}_{\mathsf{FB}}$ denote the maximizer. Notice that we put the weight $\frac{1}{k_j - 1}$ to adjust the contributions of the pairwise comparisons generated from the partial rankings over subsets with different sizes.

**Theorem 4.** *With high probability, $\|\widehat{\theta}_{\mathsf{FB}} - \theta^*\|_2 \leq 2(1 + e^{2b})^2 \frac{\sqrt{mk \log n}}{\lambda_2}$. Furthermore, suppose $S_1^m$ are chosen independently and uniformly at random among all possible subsets of $[n]$. There exists a positive constant $C > 0$ such that if $mk \geq Cn \log n$, then with high probability, $\|\widehat{\theta}_{\mathsf{FB}} - \theta^*\|_2 \leq 4(1 + e^{2b})^2 \sqrt{\frac{n^2 \log n}{mk}}$.*

Theorem 4 shows that the error rates of $\widehat{\theta}_{\mathsf{FB}}$ inversely depend on $\lambda_2$. When the comparison graph $G$ is an expander, i.e., $\lambda_2 \sim \lambda_n$, the upper bound is only larger than the Cramér-Rao lower bound by a logarithmic factor. The similar observation holds for the ML estimator as shown in Theorem 3. With the random item assignment scheme, Theorem 4 imply that the FB only need $mk = \Omega(n \log n)$ to achieve the reliable inference, which is optimal up to a $\log n$ factor in view of the oracle lower bound in Theorem 1.

**Comparison to previous work**   The rank breaking schemes considered in [8, 9] breaks the full rankings according to rank positions while our schemes break the partial rankings according to the item indices. The results in [8, 9] establish the consistency of the generalized method of moments under the rank breaking schemes when the data consists of full rankings. In contrast, Corollary 2 and Theorem 4 apply to the more general setting with partial rankings and provide the finite-sample error rates, proving the optimality of the random IB and FB with the random item assignment scheme.

## 2.5 Numerical experiments

Suppose there are $n = 1024$ items and $\theta^*$ is uniformly distributed over $[-b, b]$. We first generate $d$ full rankings over 1024 items according to the PL model with parameter $\theta^*$. Then for each fixed $k \in \{512, 256, \ldots, 2\}$, we break every full ranking $\sigma$ into $n/k$ partial rankings over subsets of size $k$ as follows: Let $\{S_j\}_{j=1}^{n/k}$ denote a partition of $[n]$ generated uniformly at random such that $S_j \cap S_{j'} = \emptyset$ for $j \neq j'$ and $|S_j| = k$ for all $j$; generate $\{\sigma_j\}_{j=1}^{n/k}$ such that $\sigma_j$ is the partial ranking over set $S_j$ consistent with $\sigma$. In this way, in total we get $m = dn/k$ $k$-way comparisons which are all independently generated from the PL model. We apply the minorization-maximization (MM) algorithm proposed in [7] to compute the ML estimator $\widehat{\theta}_{\mathsf{ML}}$ based on the $k$-way comparisons and the estimator $\widehat{\theta}_{\mathsf{FB}}$ based on the pairwise comparisons induced by the FB. The estimation error is measured by the rescaled mean square error (MSE) defined by $\log_2\left(\frac{mk}{n^2}\|\widehat{\theta} - \theta^*\|_2^2\right)$.

We run the simulation with $b = 2$ and $d = 16, 64$. The results are depicted in Fig. 1. We also plot the Cramér-Rao (CR) limit given by $\log_2\left(1 - \frac{1}{k}\sum_{l=1}^{k}\frac{1}{l}\right)^{-1}$ as per Theorem 2. The oracle lower bound in Theorem 1 implies that the rescaled MSE is at least 0. We can see that the rescaled MSE of the ML estimator $\widehat{\theta}_{\mathsf{ML}}$ is close to the CR limit and approaches the oracle lower bound as $k$ becomes large, suggesting the ML estimator is minimax-optimal. Furthermore, the rescaled MSE of $\widehat{\theta}_{\mathsf{FB}}$ under FB is approximately twice larger than the CR limit, suggesting that the FB is minimax-optimal up to a constant factor.

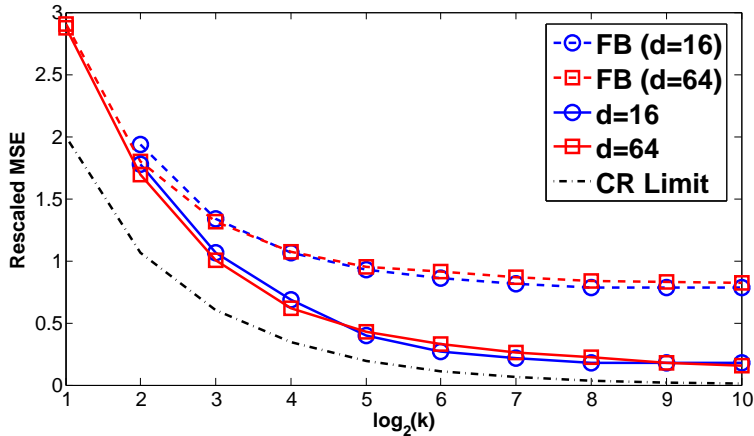

Figure 1: The error rate based on $nd/k$ $k$-way comparisons with and without full breaking.

Finally, we point out that when $d = 16$ and $\log_2(k) = 1$, the MSE returned by the MM algorithm is infinity. Such singularity occurs for the following reason. Suppose we consider a directed comparison graph with nodes corresponding to items such that for each $(i, j)$, there is a directed edge $(i \to j)$ if item $i$ is ever ranked higher than $j$. If the graph is not strongly connected, i.e., if there exists a partition of the items into two groups $A$ and $B$ such that items in $A$ are always ranked higher than items in $B$, then if all $\{\theta_i : i \in A\}$ are increased by a positive constant $a$, and all $\{\theta_i : i \in B\}$ are decreased by another positive constant $a'$ such that all $\{\theta_i, i \in [n]\}$ still sum up to zero, the log likelihood (1) must increase; thus, the log likelihood has no maximizer over the parameter space $\Theta$, and the MSE returned by the MM algorithm will diverge. Theoretically, if $b$ is a constant and $d$ exceeds the order of $\log n$, the directed comparison graph will be strongly connected with high probability and so such singularity does not occur in our numerical experiments when $d \geq 64$. In practice we can deal with this singularity issue in three ways: 1) find the strongly connected components and then run MM in each component to come up with an estimator of $\theta^*$ restricted to each component; 2) introduce a proper prior on the parameters and use Bayesian inference to come up with an estimator (see [16]); 3) add to the log likelihood objective function a regularization term based on $\|\theta\|_2$ and solve the regularized ML using the gradient descent algorithms (see [10]).

# 3 Proofs

We sketch the proof of our two upper bounds given by Theorem 3 and Theorem 4. The proofs of other results can be found in the supplementary file. We introduce some additional notations used in the proof. For a vector $x$, let $\|x\|_2$ denote the usual $l_2$ norm. Let $\mathbf{1}$ denote the all-one vector and $\mathbf{0}$ denote the all-zero vector with the appropriate dimension. Let $\mathcal{S}^n$ denote the set of $n \times n$ symmetric matrices with real-valued entries. For $X \in \mathcal{S}^n$, let $\lambda_1(X) \leq \lambda_2(X) \leq \cdots \leq \lambda_n(X)$ denote its eigenvalues sorted in increasing order. Let $\mathrm{Tr}(X) = \sum_{i=1}^n \lambda_i(X)$ denote its trace and $\|X\| = \max\{-\lambda_1(X), \lambda_n(X)\}$ denote its spectral norm. For two matrices $X, Y \in \mathcal{S}^n$, we write $X \leq Y$ if $Y - X$ is positive semi-definite, i.e., $\lambda_1(Y - X) \geq 0$. Recall that $\mathcal{L}(\theta)$ is the log likelihood function. Let $\nabla\mathcal{L}(\theta)$ denote its gradient and $H(\theta) \in \mathcal{S}^n$ denote its Hessian matrix.

## 3.1 Proof of Theorem 3

The main idea of the proof is inspired from the proof of [10, Theorem 4]. We first introduce several key auxiliary results used in the proof. Observe that $\mathbb{E}_{\theta^*}[\nabla L(\theta^*)] = 0$. The following lemma upper bounds the deviation of $\nabla L(\theta^*)$ from its mean.

**Lemma 1.** *With probability at least* $1 - \frac{2e^2}{n}$,
$$\|\nabla\mathcal{L}(\theta^*)\|_2 \leq \sqrt{2mk \log n} \tag{3}$$

Observed that $-H(\theta)$ is positive semi-definite with the smallest eigenvalue equal to zero. The following lemma lower bounds its second smallest eigenvalue.

**Lemma 2.** *Fix any* $\theta \in \Theta_b$. *Then*
$$\lambda_2\left(-H(\theta)\right) \geq \begin{cases} \frac{e^{2b}}{(1+e^{2b})^2}\lambda_2 & \textit{If } k = 2, \\ \frac{1}{4e^{4b}}\left(\lambda_2 - 16e^{2b}\sqrt{\lambda_n \log n}\right) & \textit{If } k > 2, \end{cases} \tag{4}$$
*where the inequality holds with probability at least* $1 - n^{-1}$ *in the case with* $k > 2$.

*Proof of Theorem 3.* Define $\Delta = \widehat{\theta}_{\mathsf{ML}} - \theta^*$. It follows from the definition that $\Delta$ is orthogonal to the all-one vector. By the definition of the ML estimator, $\mathcal{L}(\widehat{\theta}_{\mathsf{ML}}) \geq \mathcal{L}(\theta^*)$ and thus
$$\mathcal{L}(\widehat{\theta}_{\mathsf{ML}}) - \mathcal{L}(\theta^*) - \langle\nabla\mathcal{L}(\theta^*), \Delta\rangle \geq -\langle\nabla\mathcal{L}(\theta^*), \Delta\rangle \geq -\|\nabla\mathcal{L}(\theta^*)\|_2\|\Delta\|_2, \tag{5}$$
where the last inequality holds due to the Cauchy-Schwartz inequality. By the Taylor expansion, there exists a $\theta = a\widehat{\theta}_{\mathsf{ML}} + (1-a)\theta^*$ for some $a \in [0,1]$ such that
$$\mathcal{L}(\widehat{\theta}_{\mathsf{ML}}) - \mathcal{L}(\theta^*) - \langle\nabla\mathcal{L}(\theta^*), \Delta\rangle = \frac{1}{2}\Delta^\top H(\theta)\Delta \leq -\frac{1}{2}\lambda_2(-H(\theta))\|\Delta\|_2^2, \tag{6}$$
where the last inequality holds because the Hessian matrix $-H(\theta)$ is positive semi-definite with $H(\theta)\mathbf{1} = \mathbf{0}$ and $\Delta^\top\mathbf{1} = 0$. Combining (5) and (6),
$$\|\Delta\|_2 \leq 2\|\nabla\mathcal{L}(\theta^*)\|_2/\lambda_2(-H(\theta)). \tag{7}$$
Note that $\theta \in \Theta_b$ by definition. The theorem follows by Lemma 1 and Lemma 2. □

## 3.2 Proof of Theorem 4

It follows from the definition of $\mathcal{L}(\theta)$ given by (2) that
$$\nabla_i\mathcal{L}(\theta^*) = \sum_{j:i\in S_j} \frac{1}{k_j - 1} \sum_{i'\in S_j:i'\neq i} \left[\mathbb{I}_{\left\{\sigma_j^{-1}(i)<\sigma_j^{-1}(i')\right\}} - \frac{\exp(\theta_i^*)}{\exp(\theta_i^*) + \exp(\theta_{i'}^*)}\right], \tag{8}$$
which is a sum of $d_i$ independent random variables with mean zero and bounded by 1. By Hoeffding's inequality, $|\nabla_i L(\theta^*)| \leq \sqrt{d_i \log n}$ with probability at least $1 - 2n^{-2}$. By union bound, $\|\nabla L(\theta^*)\|_2 \leq \sqrt{mk \log n}$ with probability at least $1 - 2n^{-1}$. The Hessian matrix is given by
$$H(\theta) = -\sum_{j=1}^m \frac{1}{2(k_j-1)} \sum_{i,i'\in S_j} (e_i - e_{i'})(e_i - e_{i'})^\top \frac{\exp(\theta_i + \theta_{i'})}{[\exp(\theta_i) + \exp(\theta_{i'})]^2}.$$
If $|\theta_i| \leq b, \forall i \in [n]$, $\frac{\exp(\theta_i+\theta_{i'})}{[\exp(\theta_i)+\exp(\theta_{i'})]^2} \geq \frac{e^{2b}}{(1+e^{2b})^2}$. It follows that $-H(\theta) \geq \frac{e^{2b}}{(1+e^{2b})^2}L$ for $\theta \in \Theta_b$ and the theorem follows from (7).

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
