[Supplementary Material]

# Minimax-optimal Inference from Partial Rankings: Supplementary Material

We introduce some additional notations used in the proof. The first-order partial derivative of $\mathcal{L}(\theta)$ is given by

$$\nabla_i \mathcal{L}(\theta) = \sum_{j:i\in S_j} \sum_{\ell=1}^{k_j-1} \mathbb{I}_{\{\sigma_j^{-1}(i)\geq\ell\}} \left[ \mathbb{I}_{\{\sigma_j(\ell)=i\}} - \frac{\exp(\theta_i)}{\exp(\theta_{\sigma_j(\ell)}) + \cdots + \exp(\theta_{\sigma_j(k_j)})} \right], \forall i \in [n] \quad (1)$$

and the Hessian matrix $H(\theta) \in \mathcal{S}^n$ with $H_{ii'}(\theta) = \frac{\partial^2 \mathcal{L}(\theta)}{\partial\theta_i\partial\theta_{i'}}$ is given by

$$H(\theta) = -\frac{1}{2} \sum_{j=1}^{m} \sum_{i,i'\in S_j} (e_i - e_{i'})(e_i - e_{i'})^\top \sum_{\ell=1}^{k_j-1} \frac{\exp(\theta_i + \theta_{i'})\mathbb{I}_{\{\sigma_j^{-1}(i),\sigma_j^{-1}(i')\geq\ell\}}}{[\exp(\theta_{\sigma_j(\ell)}) + \cdots + \exp(\theta_{\sigma_j(k_j)})]^2}. \quad (2)$$

It follows from the definition that $-H(\theta)$ is positive semi-definite for any $\theta \in \mathbb{R}^n$. Define $L_j \in S^n$ as

$$L_j = \frac{1}{2(k_j - 1)} \sum_{i,i'\in S_j} (e_i - e_{i'})(e_i - e_{i'})^\top,$$

and then the Laplacian of the pairwise comparison graph $G$ satisfies $L = \sum_{j=1}^{m} L_j$.

## 1 Proof of Theorem 1

We first introduce a key auxiliary result used in the proof. Let $F$ be a fixed CDF (to be used in the Thurstone model), let $b > 0$ and suppose $\theta$ is a parameter to be estimated with $\theta \in [-b, b]$ from observation $U = (U_1, \ldots, U_d)$, where the $U_i$'s are independent with the common CDF given by $F(c - \theta)$. The following proposition gives a lower bound on the average MSE for a fixed prior distribution based on Van Trees inequality [1].

**Proposition 1.** *Let $p_0$ be a probability density on $[-1, 1]$ such that $p_0(1) = p_0(-1) = 0$ and define the prior density of $\Theta$ as $p(\theta) = \frac{1}{b}p_0(\frac{\theta}{b})$. Then for any estimator $T(U)$ of $\Theta$,*

$$E[(\Theta - T(U))^2] \geq \frac{1}{d}\frac{1}{I(\mu) + I(p_0)/(b^2 d)},$$

*where $\mu$ is the probability density function of $F$ with $I(\mu) = \int \frac{(\mu'(x))^2}{\mu(x)}dx$ and $I(p_0) = \int_{-1}^{1} \frac{(p_0'(\theta))^2}{p_0(\theta)}d\theta$.*

*Proof.* It follows from the Van Trees inequality that

$$E[(\Theta - T(U))^2] \geq \frac{1}{\int I(\theta)p(\theta)d\theta + I(p)},$$

where the Fisher information $I(\theta) = dI(\mu)$ and

$$I(p) = \int_{-b}^{b} \frac{(p'(\theta))^2}{p(\theta)}d\theta = \frac{1}{b^2}\int_{-1}^{1} \frac{(p_0'(\theta))^2}{p_0(\theta)}d\theta = \frac{1}{b^2}I(p_0).$$

$\square$

*Proof of Theorem 1.* Let $\widehat{\theta}$ be a given estimator. The minimax MSE for $\widehat{\theta}$ is greater than or equal to the average MSE for a given prior distribution on $\theta^*$. Let $p_0(\theta) = \cos^2(\pi\theta/2)$, then $I(p_0) = \pi^2$. Define $p(\theta) = \frac{1}{b}p_0(\frac{\theta}{b})$. If $n$ is even we use the following prior distribution. The prior distribution of $\theta_i^*$ for $i$ odd is $p(\theta)$ and for $i$ even, $\theta_i^* \equiv -\theta_{i-1}^*$. If $n$ is odd use the same distribution for $\theta_1^*$ through $\theta_{n-1}^*$ and set $\theta_n^* \equiv 0$. Note that $\theta^* \in \Theta_b$ with probability one. For simplicity, we assume $n$ is odd in the rest of this proof; the modification for $n$ even is trivial. We use the genie argument, so that the observer can see the hidden utilities in the Thurstone model. The estimation of $\theta^*$ decouples into $\lfloor \frac{n}{2} \rfloor$ disjoint problems, so we can focus on the estimation of $\theta_1$ from the vector of random variables $U = (U_1, \ldots, U_{d_1})$ associated with item 1 and the vector of random variables $V = (V_1, \ldots, V_{d_2})$ associated with item 2. The distribution functions of the $U_i$'s are all $F(c - \theta_1^*)$ and the distribution functions of the $V_i$'s are all $F(c + \theta_1^*)$, and the $U$'s and $V$'s are all mutually independent given $\theta^*$. Recall that $\mu$ is the probability density function of $F$, i.e., $\mu = F'$. The Fisher information for each of the $d_1 + d_2$ observations is $I(\mu)$, so that Proposition 1 carries over to this situation with $d = d_1 + d_2$. Therefore, for any estimator $T(U, V)$ of $\Theta_1^*$ (the random version of $\theta_1^*$),

$$E[(\Theta_1^* - T(U,V))^2] \geq \frac{1}{d_1 + d_2} \frac{1}{I(\mu) + \pi^2/(b^2(d_1 + d_2))}$$

By this reasoning, for any odd value of $i$ with $1 \leq i < n$ we have

$$E[(\widehat{\theta}_i - \theta_i^*)^2] + E[(\widehat{\theta}_{i+1} - \theta_{i+1}^*)^2] \geq \frac{2}{I(\mu) + \pi^2/(b^2(d_1 + d_2))} \frac{1}{d_i + d_{i+1}}$$

$$\geq \frac{1}{2I(\mu) + 2\pi^2/(b^2(d_1 + d_2))} \left( \frac{1}{d_{i+1}} + \frac{1}{d_{i+2}} \right).$$

Summing over all odd values of $i$ in the range $1 \leq i < n$ yields the theorem. Furthermore, since $\sum_{i=1}^n d_i = mk$, by Jensen's inequality, $\sum_{i=2}^n \frac{1}{d_i} \geq \frac{(n-1)^2}{\sum_{i=2}^n d_i} \geq \frac{(n-1)^2}{mk}$. □

## 2 Proof of Theorem 2

The Fisher information matrix is defined as $I(\theta) = -\mathbb{E}_\theta[H(\theta)]$ and given by

$$I(\theta) = \frac{1}{2} \sum_{j=1}^m \sum_{i,i' \in S_j} (e_i - e_{i'})(e_i - e_{i'})^\top \sum_{l=1}^{k_j - 1} \mathbb{P}_\theta[\sigma_j^{-1}(i), \sigma_j^{-1}(i') \geq \ell] \frac{e^{\theta_i + \theta_{i'}}}{[e^{\theta_{\sigma_j(\ell)}} + \cdots + e^{\theta_{\sigma_j(k_j)}}]^2}.$$

Since $-H(\theta)$ is positive semi-definite, it follows that $I(\theta)$ is positive semi-definite. Moreover, $\lambda_1(I(\theta))$ is zero and the corresponding eigenvector is the normalized all-one vector. Fix any unbiased estimator $\widehat{\theta}$ of $\theta \in \Theta_b$. Since $\widehat{\theta} \in \mathcal{U}$, $\widehat{\theta} - \theta$ is orthogonal to $\mathbf{1}$. The Cramér-Rao lower bound then implies that $\mathbb{E}[\|\widehat{\theta} - \theta\|^2] \geq \sum_{i=2}^n \frac{1}{\lambda_i(I(\theta))}$. Taking the supremum over both sides gives

$$\sup_\theta \mathbb{E}[\|\widehat{\theta} - \theta\|^2] \geq \sup_\theta \sum_{i=2}^n \frac{1}{\lambda_i(I(\theta))} \geq \sum_{i=2}^n \frac{1}{\lambda_i(I(0))}.$$

If $\theta$ equals the all-zero vector, then

$$\mathbb{P}[\sigma_j^{-1}(i), \sigma_j^{-1}(i') \geq \ell] = \frac{(k_j - 2)(k_j - 3)\cdots(k_j - \ell)}{k_j(k_j - 1)\cdots(k_j - \ell + 2)} = \frac{(k_j - \ell + 1)(k_j - \ell)}{k_j(k_j - 1)}.$$

It follows from the definition that

$$I(0) = \frac{1}{2} \sum_{j=1}^m \sum_{i,i' \in S_j} (e_i - e_{i'})(e_i - e_{i'})^\top \sum_{l=1}^{k_j - 1} \frac{k_j - \ell}{k_j(k_j - 1)(k_j - \ell + 1)} \leq \left( 1 - \frac{1}{k_{\max}} \sum_{\ell=1}^{k_{\max}} \frac{1}{\ell} \right) L.$$

By Jensen's inequality,

$$\sum_{i=2}^n \frac{1}{\lambda_i} \geq \frac{(n-1)^2}{\sum_{i=2}^n \lambda_i} = \frac{(n-1)^2}{\text{Tr}(L)} = \frac{(n-1)^2}{\sum_{i=1}^n d_i} = \frac{(n-1)^2}{mk}.$$

# 3 Proof of Lemma 1

The idea of the proof is to view $\nabla\mathcal{L}(\theta^*)$ as the final value of a discrete time vector-valued martingale with values in $\mathbb{R}^n$. Consider a user that ranks items $1, \ldots, k$. The PL model for the ranking can be generated in a series of $k-1$ rounds. In the first round, the top rated item for the user is found. Suppose it is item $I$. This contributes the term $e_I - (p_1, p_2, \ldots, p_k, 0, 0, \ldots, 0)$ to $\nabla\mathcal{L}(\theta^*)$, where $p_i = P\{I = i\}$. This contribution is a mean zero random vector in $\mathbb{R}^n$ and its norm is less than one. For notational convenience, suppose $I = k$. In the second round, item $k$ is removed from the competition, and an item $J$ is to be selected at random from among $\{1, \ldots, k-1\}$. If $q_j$ denotes $P\{J = j\}$ for $1 \le j \le k-1$, then the contribution of the second round for the user to $\nabla\mathcal{L}(\theta^*)$ is the random vector $e_J - (q_1, q_2, \ldots, q_{k-1}, 0, 0, \ldots, 0)$, which has conditional mean zero (given $I$) and norm less than or equal to one. Considering all $m$ users and $k_j - 1$ rounds for user $j$, we see that $\nabla\mathcal{L}(\theta^*)$ is the value of a discrete-time martingale at time $m(k-1)$ such that the martingale has initial value zero and increments with norm bounded by one. By the vector version of the Azuma-Hoeffding inequality found in [2, Theorem 1.8] we have

$$\mathbb{P}\{\|\nabla\mathcal{L}(\theta^*)\| \ge \delta\} \le 2e^2 e^{-\frac{\delta^2}{2m(k-1)}},$$

which implies the result.

# 4 Proof of Lemma 2

We first introduce a key auxiliary result used in the proof.

**Claim 1.** *Given $\theta \in \mathbb{R}^r$, let $A = diag(p) - pp^T$, where $p$ is the column probability vector with $p_i = e^{\theta_i}/(e^{\theta_1} + \cdots + e^{\theta_r})$ for each $i$. If $|\theta_i| \le b$, for $1 \le i \le r$, then $\lambda_2(A) \ge \frac{1}{re^{2b}}$. Equivalently, $e^{2b}A \ge B$ where $B = \frac{1}{r}diag(\mathbf{1}) - \frac{1}{r^2}\mathbf{11}^\top$.*

*Proof.* Fix $\theta$ satisfying the conditions of the lemma. It is easy to see that for each $i$, $p_i \ge \frac{1}{re^{2b}}$. The matrix $A$ is positive semidefinite, and its smallest eigenvalue is zero, with the corresponding eigenvector $\mathbf{1}$. So $\lambda_2(A) = \min_\alpha \alpha^T A\alpha$ subject to the constraints $\alpha^T \mathbf{1} = 0$ and $\|\alpha\|^2 = 1$. For $\alpha$ satisfying the constraints,

$$
\begin{aligned}
\alpha^T A\alpha &= \sum_i \alpha_i^2 p_i - \left(\sum_j \alpha_j p_j\right)^2 = \sum_i \left(\alpha_i - \sum_j \alpha_j p_j\right)^2 p_i \\
&= \min_c \sum_{i=1}^r (\alpha_i - c)^2 p_i \ge \min_c \sum_{i=1}^r (\alpha_i - c)^2 \frac{1}{re^{2b}} \\
&= \sum_{i=1}^r \alpha_i^2 \frac{1}{re^{2b}} = \frac{1}{re^{2b}}
\end{aligned}
$$

The proof of the first part of the lemma is complete. We remark that the bound of the lemma is nearly tight for the case $\theta_1 = \ldots = \theta_{r-1} = b$ and $\theta_r = -b$, for which $\lambda_2(A) = \frac{e^{2b}r}{((r-1)e^{2b}+1)^2}$. The final equivalence mentioned in the lemma follows from the facts $\lambda_1(e^{2b}A) = \lambda_1(B) = 0$ with common corresponding eigenvector $\mathbf{1}$, and $\lambda_i(e^{2b}A) \ge \frac{1}{r} = \lambda_i(B)$ for $2 \le i \le r$. $\square$

*Proof of Lemma 2.* **Case $k_j = 2, \forall j \in [m]$:** The Hessian matrix simplifies as

$$H(\theta) = -\frac{1}{2}\sum_{j=1}^m \sum_{i,i'\in S_j} (e_i - e_{i'})(e_i - e_{i'})^\top \frac{\exp(\theta_i)}{\exp(\theta_i) + \exp(\theta_{i'})} \frac{\exp(\theta_{i'})}{\exp(\theta_i) + \exp(\theta_{i'})}.$$

Observe that $H(\theta)$ is deterministic given $S_1^m$. Since $|\theta_i| \le b, \forall i \in [n]$,

$$\frac{\exp(\theta_i)\exp(\theta_{i'})}{[\exp(\theta_i) + \exp(\theta_{i'})]^2} \ge \frac{e^{2b}}{(1 + e^{2b})^2}.$$

It follows that $-H(\theta) \geq \frac{e^{2b}}{(1+e^{2b})^2} L$ and the theorem follows.

**Case $k_j > 2$ for some $j \in [m]$:** The Hessian matrix $H(\theta)$ depends on $\sigma_1^m$ and therefore is random given $S_1^m$. For a given user $j$, and $\ell$ with $1 \leq \ell \leq k_j - 1$, let $S^{(j,\ell)}$ denote the set of items contending for the $\ell^{th}$ position in the ranking of user $j$ after higher ranking items have been selected: $S^{(j,\ell)} = \{i : \sigma_j^{-1}(i) \geq \ell\}$, let $\mathbf{1}^{(j,\ell)}$ denote the indicator vector for the set $S^{(j,\ell)}$, and let $p^{(j,\ell)}$ denote the corresponding probability column vector for the selection:

$$p_i^{(j,\ell)} = P(\sigma_j(\ell) = i | \sigma_j(1), \dots, \sigma_j(\ell-1)) = \frac{\mathbf{1}_i^{(j,\ell)} e^{\theta_i}}{\sum_{i' \in S_{j,\ell}} e^{\theta_{i'}}}$$

The Hessian can be written as $H(\theta) = \sum_{j=1}^m \sum_{\ell=1}^{k_j-1} H^{(j,\ell)}$ where

$$-H^{(j,\ell)} = \frac{1}{2} \sum_{i,i' \in S^{(j,\ell)}} (e_i - e_{i'})(e_i - e_{i'})^\top p_i^{(j,\ell)} p_{i'}^{(j,\ell)} = \mathrm{diag}(p^{(j,\ell)}) - p^{(j,\ell)}(p^{(j,\ell)})^\top$$

By Claim 1 applied to the restriction of $-H^{(j,\ell)}$ to $S^{(j,\ell)} \times S^{(j,\ell)}$,

$$
\begin{aligned}
-e^{2b} H^{(j,\ell)} &\geq \frac{1}{k_j - \ell + 1} \mathrm{diag}(\mathbf{1}^{(j,\ell)}) - \frac{1}{(k_j - \ell + 1)^2} \mathbf{1}^{(j,\ell)}(\mathbf{1}^{(j,\ell)})^\top \\
&= \frac{1}{2(k_j - \ell + 1)^2} \sum_{i,i' \in S^{(j,\ell)}} (e_i - e_{i'})(e_i - e_{i'})^\top
\end{aligned}
\tag{3}
$$

Summing over $j$ and $\ell$ in (3) and noting that $k_j - \ell + 1 \leq k_j$ for all $j, \ell$ yields

$$-e^{2b} H(\theta) \geq \frac{1}{2} \sum_{j=1}^m \sum_{i,i' \in S_j} (e_i - e_{i'})(e_i - e_{i'})^\top \frac{1}{k_j^2} \sum_{\ell=1}^{k_j-1} \mathbb{I}_{\{\sigma_j^{-1}(i), \sigma_j^{-1}(i') \geq \ell\}} := \tilde{L} \tag{4}$$

Observe that

$$\sum_{\ell=1}^{k_j-1} \mathbb{P}_\theta\left[\sigma_j^{-1}(i), \sigma_j^{-1}(i') \geq \ell\right] = 1 + \sum_{i'' \in S_j} \mathbb{I}_{\{i'' \neq i, i'\}} \frac{e^{\theta_{i''}}}{e^{\theta_i} + e^{\theta_{i'}} + e^{\theta_{i''}}} \geq 1 + \frac{k_j - 2}{2e^{2b} + 1} \geq \frac{k_j + 1}{3e^{2b}}.$$

Recall that $L$ is the Laplacian of $G$ and $L = \sum_{j=1}^m L_j$. It follows that

$$
\begin{aligned}
\mathbb{E}_\theta[\tilde{L}] &= \frac{1}{2} \sum_{j=1}^m \sum_{i,i' \in S_j} (e_i - e_{i'})(e_i - e_{i'})^\top \frac{1}{k_j^2} \sum_{\ell=1}^{k_j-1} \mathbb{P}_\theta[\sigma_j^{-1}(i), \sigma_j^{-1}(i') \geq \ell] \\
&\geq \frac{1}{2} \sum_{j=1}^m \sum_{i,i' \in S_j} (e_i - e_{i'})(e_i - e_{i'})^\top \frac{k_j + 1}{3e^{2b} k_j^2} \\
&\geq \frac{1}{2} \sum_{j=1}^m \sum_{i,i' \in S_j} (e_i - e_{i'})(e_i - e_{i'})^\top \frac{1}{4e^{2b}(k_j - 1)} = \frac{1}{4e^{2b}} L
\end{aligned}
\tag{5}
$$

Define $a_{ii'} = \frac{1}{k_j^2} \sum_{\ell=1}^{k_j-1} \left( \mathbb{I}_{\{\sigma_j^{-1}(i), \sigma_j^{-1}(i') \geq \ell\}} - \mathbb{P}_\theta[\sigma_j^{-1}(i), \sigma_j^{-1}(i') \geq \ell] \right)$. Then

$$\tilde{L} - \mathbb{E}_\theta[\tilde{L}] = \frac{1}{2} \sum_{j=1}^m \left( \sum_{i,i' \in S_j} a_{ii'}(e_i - e_{i'})(e_i - e_{i'})^\top \right) := \sum_{j=1}^m Y_j.$$

Observe that $|a_{ii'}| \leq \frac{1}{k_j}$ and therefore $-\frac{(k_j-1)}{k_j} L_j \leq Y_j \leq \frac{(k_j-1)}{k_j} L_j$. Furthermore, $\|L_j\| = \frac{k_j}{k_j - 1}$ and thus $\|Y_j\| \leq 1$. Moreover, $Y_j^2 = \sum_{i,i',i'' \in S_j} a_{ii'} a_{ii''}(e_i - e_{i'})(e_i - e_{i''})^\top$. It follows that for any vector $x \in \mathbb{R}^n$,

$$
\begin{aligned}
x^\top Y_j^2 x &= \sum_{i,i',i'' \in S_j} a_{ii'} a_{ii''}(x_i - x_{i'})(x_i - x_{i''}) \leq \frac{1}{k_j^2} \sum_{i,i',i'' \in S_j} |x_i - x_{i'}| |x_i - x_{i''}| \\
&= \frac{1}{k_j^2} \sum_{i \in S_j} \left( \sum_{i' \in S_j} |x_i - x_{i'}| \right)^2 \leq \frac{1}{k_j} \sum_{i,i' \in S_j} (x_i - x_{i'})^2 = 2x^\top L_j x,
\end{aligned}
$$

where the last inequality follows from the Cauchy-Swartz inequality. Therefore, $Y_j^2 \leq 2L_j$. It follows that $\sum_{j=1}^m \mathbb{E}_\theta[Y_j^2] \leq 2L$ and thus $\|\sum_{j=1}^m \mathbb{E}_\theta[Y_j^2]\| \leq 2\lambda_n$. By the matrix Bernstein inequality [3], with probability at least $1 - n^{-1}$,

$$\|\tilde{L} - \mathbb{E}_\theta[\tilde{L}]\| \leq 2\sqrt{\lambda_n \log n} + \frac{2}{3}\log n.$$

By the assumption that $\lambda_n \geq C \log n$ for some sufficiently large constant $C$, $\|\tilde{L} - \mathbb{E}_\theta[\tilde{L}]\| \leq 4\sqrt{\lambda_n \log n}$. It follows from (4) and (5) that

$$\lambda_2(-H(\theta)) \geq \frac{1}{e^{2b}}\lambda_2(\tilde{L}) \geq \frac{1}{e^{2b}}\left(\frac{1}{4e^{2b}}\lambda_2 - 4\sqrt{\lambda_n \log n}\right).$$

$\square$

## 5  Proof of Corollary 1

Recall that $L = \sum_{j=1}^m L_j$. Observe that $\mathbb{E}[L_j] = \frac{k_j}{n-1}\left(I - \frac{1}{n}\mathbf{1}\mathbf{1}^\top\right)$. Define $Z_j = L_j - \mathbb{E}[L_j]$. Then $Z_1, \ldots, Z_m$ are independent symmetric random matrices with zero mean. Note that

$$\|Z_j\| \leq \|L_j\| + \|\mathbb{E}[L_j]\| \leq \frac{k_j}{k_j - 1} + \frac{k_j}{n-1} \leq 4.$$

Moreover,

$$\mathbb{E}[Z_j^2] = \frac{k_j^2}{(k_j-1)(n-1)}\left(I - \frac{1}{n}\mathbf{1}\mathbf{1}^\top\right) - \frac{k_j^2}{(n-1)^2}\left(I - \frac{1}{n}\mathbf{1}\mathbf{1}^\top\right).$$

Therefore, $\|\sum_{j=1}^m \mathbb{E}[Z_j^2]\| \leq \frac{2mk}{n-1}$. By the matrix Bernstein inequality [3], with probability at least $1 - n^{-1}$,

$$\|L - \mathbb{E}[L]\| \leq 2\sqrt{\frac{mk \log n}{n-1}} + \frac{8}{3}\log n \leq 4\sqrt{\frac{mk \log n}{n-1}} \leq \frac{mk}{2(n-1)}.$$

where the last two inequalities follow from the assumption that $mk \geq C \log n$ for some sufficiently large constant $C$. Since $\mathbb{E}[L] = \frac{mk}{n-1}\left(I - \frac{1}{n}\mathbf{1}\mathbf{1}^\top\right)$, the smallest eigenvalue of $\mathbb{E}[L]$ is zero and all the other eigenvalues equal $\frac{mk}{n-1}$. It follows that

$$|\lambda_i - \frac{mk}{n-1}| \leq \|L - \mathbb{E}[L]\| \leq \frac{mk}{2(n-1)}, \quad 2 \leq i \leq n,$$

and thus $\lambda_2 \geq \frac{mk}{2(n-1)}$ and $\lambda_n \leq \frac{3mk}{2(n-1)}$. By the assumption that $mk \geq Ce^{2b}\log n$ for some sufficiently large constant $C$, $\lambda_2 - 16e^{2b}\sqrt{\lambda_n \log n} \geq \frac{mk}{4n}$. Then the corollary follow from Theorem 3.

## 6  Proof of Corollary 2

Without loss of generality, assume $k_j$ is even for all $j \in [m]$. After the random IB, there are $mk/2$ independent pairwise comparisons and let $L$ denote the Laplacian of the comparison graph after the breaking. Recall that $L = \sum_{j=1}^m L_j$. With random IB, we have $\mathbb{E}[L_j] = \frac{k_j}{n-1}\left(I - \frac{1}{n}\mathbf{1}\mathbf{1}^\top\right)$. Define $Z_j = L_j - \mathbb{E}[L_j]$. Then $Z_1, \ldots, Z_m$ are independent symmetric random matrices with zero mean. Moreover,

$$\|Z_j\| \leq \|L_j\| + \|\mathbb{E}[L_j]\| \leq 2 + \frac{k_j}{n-1} \leq 4,$$

and

$$\mathbb{E}[Z_j^2] = \frac{2k_j}{n-1}\left(I - \frac{1}{n}\mathbf{1}\mathbf{1}^\top\right) - \frac{k_j^2}{(n-1)^2}\left(I - \frac{1}{n}\mathbf{1}\mathbf{1}^\top\right).$$

Therefore, $\|\sum_{j=1}^m \mathbb{E}[Z_j^2]\| \leq \frac{2mk}{n-1}$. Following the same argument for proving Corollary 1, we can show that $\lambda_2(L_{\mathsf{IB}}) \geq \frac{mk}{2(n-1)}$ and the corollary follows by Theorem 3 with $k = 2$.