[Reviews · NeurIPS 2014]

Submitted by Assigned_Reviewer_9

This paper studies the rank aggregation problem where a global ranking is inferred from multiple partial rankings. While assuming the partial rankings are generated according to the Plackett-Luce (PL) model, some of the results in the paper apply to the more general Thurstone’s model as well. It provides theoretical results quantifying the required number of item assignments from users and analyzes the case where only pairwise comparisons are used as aggregation input. I find the results of the latter, i.e., rank-breaking upper bounds, especially interesting. At the end of the paper, the theoretical bounds are empirically tested with synthetic data.

The paper is well written and well organized. I have only two technical questions/remarks.

To apply the MM (minorization-maximization) algorithm proposed by Hunter, one has to make sure there is no dominating or dominated item in the set. More precisely, we have to assume in every possible partition of the items into two nonempty subsets, some item in one set beats some item in the other set at least once. How does this assumption hold in Section 2.5?

As the authors mention the Netflix data Section 1 as a motivating example, I am curious if there is any empirical result related to that data.
Summary: This paper studies the rank aggregation problem where the input partial rankings are modeled with Plackett-Luce model. Interesting paper with nice theoretical results.

Submitted by Assigned_Reviewer_11

This work studies the use of the Plackett-Luce model on the problem of rank aggregation with partial feedbacks. It mainly consists in a theoretical study of the estimation error depending on the quantity of feedbacks that are given by the users and depending on the assignments of the items to the users. The authors provide first two lower bounds on the estimation error of the PL model – one for any estimator, the other (tighter) only for unbiased estimators. Then, the authors present three different upper bounds, depending on the how the feedback is observed (partial ranking or pairwise preferences), with two schemes for the observations of the pairs.
I liked the technical quality of this theoretical analysis (even if I didn’t checked the proof deeply) for several reasons:
- The results proved are really non-trivial,
- the authors paid attention to enlarge the scope of their results when they could – e.g. the first lower bound holds for a larger family of models than PL,
- the authors only narrowed the scope to obtain tighter bound and actually provided some insights to compare their different bounds.
- The discussion is not restricted on the pure performance of the estimation w.r.t. the quantity of feedback, but also with the way the feedback are assigned (even if I’m not familiar with this setting).
Beyond these aspects, truth should be said, the major drawback of this paper concerns its
form. The writing and the structure (1 and ½ page of notations and definitions at the beginning) all contribute to make it extremely difficult to read.
Summary: I don’t think this theoretical study is of big impact for practitioners (even if I can’t really tell about the assignment items-users part), but the theoretical results and their technicality are impressive. The formatting is to improve.

Submitted by Assigned_Reviewer_24

This paper studies the problem of rank aggregation. The authors give some theoretical guides on how to optimally assign items to user for ranking and how many item assignments are needed to achieve a target estimation errors.

[Pros]
1. The motivation is clear and the problem studied is important.
2. The theoretical results are strong and interesting.
3. The numerical experiments validates the theoretical findings.

[Cons]
1. Comparisons on different assignment are missing. The authors propose to answer the question of how to optimally assign items to users. However, the paper only discussed about the case of random assignment. If a comparison or discussion on other assignment is added, I think it would be better and interesting.
2. In line 218, is "E_{\theta}[\hat{\theta}]=\theta,\forall \theta\in \Theta_b" should be "E_{\theta}[\hat{\theta}]=\theta^*"? If not, can you give some explanation why you require it is unbiased for all \theta\in\Theta_b?
Summary: The paper is solid and the theoretical findings provide some guarantee to the random assignment in rank aggregation with partial ranking.
Author Feedback
Author rebuttal: Thank you for your thorough and constructive reviews which are of great help for us to improve the quality of the paper.

First review (Reviewer 11)
1) In section 1.1, we just introduced enough notation and definitions to precisely present the model, summarize the main results, and discuss the related work. While we tried hard to make the notation and definitions as simple as possible, we will redouble our efforts for the final version.

2) Our theoretical study provides two design insights for practitioners. First, our results show that the estimation errors inversely depend on the spectral gap of the comparison graph which is constructed from the assignment scheme. Therefore, to reduce the estimation error, practitioners can apply assignment schemes which have a large spectral gap, such as the random assignment scheme. Second, we show that the two rank-breaking schemes are nearly optimal in terms of the sample complexity, giving some justification to the use of rank-breaking schemes in practice.

Second review (Reviewer 24)
1) Theoretically, we discussed the comparisons of different assignment schemes. Our lower and upper bounds show the estimation errors under any given assignment scheme inversely depend on the spectral gap of the comparison graph, which is constructed from the assignment scheme. Therefore an assignment scheme with a larger spectral gap will have a smaller estimation error. Since the random assignment scheme has the largest spectral gap up to constants, it is order-optimal. Empirically, we only tested the performance of the random assignment scheme based on the synthetic data, and compared to the Cramer-Rao limit and the oracle lower bound, showing the random assignment scheme is indeed order-optimal. While we could also evaluate the performance of other assignment schemes arising in real datasets as well, we did not include such evaluation in the paper; the focus is to prove the minimax-optimal inference scheme and there is a page constraint.

2) Yes, the way it is written might be confusing. Here is a hopefully less confusing explanation: \hat{\theta} is an unbiased estimator of \theta^\ast if and only if E[\hat{\theta} | \theta^\ast=\theta]=\theta, for all theta \in \Theta_b. This is the usual definition of the unbiased estimator. Note that the true parameter \theta^\ast may take any value in \Theta_b. Although our Cramer-Rao lower bound only holds for the unbiased estimator, our oracle lower bound holds for any estimator.

Third review (Reviewer 9)
1) Thanks for pointing it out. Suppose we consider a directed comparison graph with nodes corresponding to items and the directed edge (i,j) denoting item i is ever ranked higher than j. If the graph is not strongly connected, i.e., if there exists a partition of the items such that items in one set always beat items in the other set, the mean square error produced by the MM algorithm will diverge. This indeed happened in our numerical experiment in Section 2.5. AS shown in the right plot in Figure 1: If d=16 (every item is compared 16 times) and k=2 (the comparison is pairwise), the MSE is infinity and that is why there is no circle marker on the curve corresponding to d=16. Theoretically, if b (dynamic range of theta) is a constant and d exceeds the order of log n, the directed comparison graph will be strongly connected with high probability and so it is not surprising that such singularity did not occur in our numerical experiments when d>=32. We will clarify this point in the final version of the paper. Note that we can deal with this singularity issue in three ways: 1) find the strongly connected components and then run MM in each component to come up with an estimator of the theta vector restricted to each component; 2)introduce a proper prior on the parameters and use Bayesian inference to come up with an estimator of the theta vector (see [Guiver and Snelson 2009]); 3) add to the log likelihood objective function a regularization term based on the L_2 norm of the theta vector and solve the regularized MLE using the gradient descent algorithms (see [Negahban-Oh-Shah 2012]). The performance evaluation of these methods for actual data is interesting but is beyond the scope of our paper.

2) Some empirical results on fitting the PL model on movie scores data have been reported previously. For example, [Guiver and Snelson 2009] tried to estimate the rankings of movie genres by fitting the PL model on the MovieLens data set which consists of 100,000 ratings (1-5) from 943 users on 1682 movies. They obtained the ranking data by turning the ratings into rankings of movie genres and then running a Bayesian inference algorithm to come up with an estimate of the theta vector of all movie genres. In a similar manner, we could turn the ratings in the Netflix data into comparisons (if there is a tie, we can break it randomly). Then we could check for the strong connectivity of the directed comparison graph. Finally, we could try running MM on each strongly connected component or solving the regularized MLE to come up with an estimate of the theta vector. This is an interesting direction which we would like to pursue in the future.